# WaveGrad: Estimating Gradients for Waveform Generation

**Nanxin Chen**[*]
Johns Hopkins University, Center for Language and Speech Processing
`bobchennan@jhu.edu`

**Yu Zhang**[†], **Heiga Zen, Ron J. Weiss, Mohammad Norouzi, William Chan**[†]
Google Research, Brain Team
`{ngyuzh,heigazen,ronw,mnorouzi,williamchan}@google.com`

## Abstract

This paper introduces *WaveGrad*, a conditional model for waveform generation which estimates gradients of the data density. The model is built on prior work on score matching and diffusion probabilistic models. It starts from a Gaussian white noise signal and iteratively refines the signal via a gradient-based sampler conditioned on the mel-spectrogram. WaveGrad offers a natural way to trade inference speed for sample quality by adjusting the number of refinement steps, and bridges the gap between non-autoregressive and autoregressive models in terms of audio quality. We find that it can generate high fidelity audio samples using as few as six iterations. Experiments reveal WaveGrad to generate high fidelity audio, outperforming adversarial non-autoregressive baselines and matching a strong likelihood-based autoregressive baseline using fewer sequential operations. Audio samples are available at `https://wavegrad.github.io/`.

## 1 Introduction

Deep generative models have revolutionized speech synthesis (Oord et al., 2016; Sotelo et al., 2017; Wang et al., 2017; Biadsy et al., 2019; Jia et al., 2019; Vasquez & Lewis, 2019). Autoregressive models, in particular, have been popular for raw audio generation thanks to their tractable likelihoods, simple inference procedures, and high fidelity samples (Oord et al., 2016; Mehri et al., 2017; Kalchbrenner et al., 2018; Song et al., 2019; Valin & Skoglund, 2019). However, autoregressive models require a large number of sequential computations to generate an audio sample. This makes it challenging to deploy them in real-world applications where faster than real time generation is essential, such as digital voice assistants on smart speakers, even using specialized hardware.

There has been a plethora of research into non-autoregressive models for audio generation, including normalizing flows such as inverse autoregressive flows (Oord et al., 2018; Ping et al., 2019), generative flows (Prenger et al., 2019; Kim et al., 2019), and continuous normalizing flows (Kim et al., 2020; Wu & Ling, 2020), implicit generative models such as generative adversarial networks (GAN) (Donahue et al., 2018; Engel et al., 2019; Kumar et al., 2019; Yamamoto et al., 2020; Bińkowski et al., 2020; Yang et al., 2020a;b; McCarthy & Ahmed, 2020) and energy score (Gritsenko et al., 2020), variational auto-encoder models (Peng et al., 2020), as well as models inspired by digital signal processing (Ai & Ling, 2020; Engel et al., 2020), and the speech production mechanism (Juvela et al., 2019; Wang et al., 2020). Although such models improve inference speed by requiring fewer sequential operations, they often yield lower quality samples than autoregressive models.

This paper introduces *WaveGrad*, a conditional generative model of waveform samples that estimates the gradients of the data log-density as opposed to the density itself. WaveGrad is simple to train, and implicitly optimizes for the weighted variational lower-bound of the log-likelihood.

---

[*]Work done during an internship at Google Brain.
[†]Equal contribution.

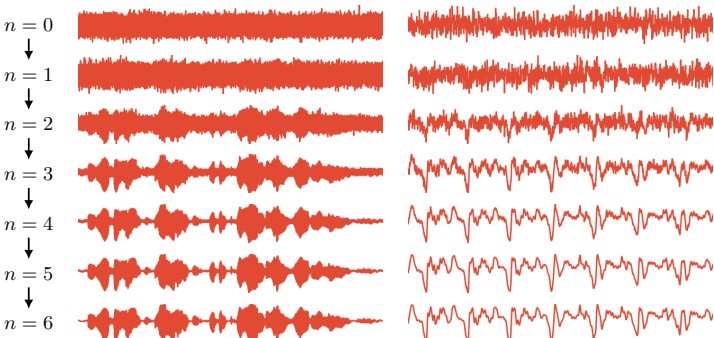

Figure 1: A visualization of the WaveGrad inference process. Starting from Gaussian noise ($n = 0$), gradient-based sampling is applied using as few as 6 iterations to achieve high fidelity audio ($n = 6$). Left: signal after each step of a gradient-based sampler. Right: zoomed view of a 50 ms segment.

WaveGrad is non-autoregressive, and requires only a constant number of generation steps during inference. Figure 1 visualizes the inference process of WaveGrad.

WaveGrad builds on a class of generative models that emerges through learning the gradient of the data log-density, also known as the Stein score function (Hyvärinen, 2005; Vincent, 2011). During inference, one can rely on the gradient estimate of the data log-density and use gradient-based samplers (e.g., Langevin dynamics) to sample from the model (Song & Ermon, 2019). Promising results have been achieved on image synthesis (Song & Ermon, 2019; 2020) and shape generation (Cai et al., 2020). Closely related are diffusion probabilistic models (Sohl-Dickstein et al., 2015), which capture the output distribution through a Markov chain of latent variables. Although these models do not offer tractable likelihoods, one can optimize a (weighted) variational lower-bound on the log-likelihood. The training objective can be reparameterized to resemble deonising score matching (Vincent, 2011), and can be interpreted as estimating the data log-density gradients. The model is non-autoregressive during inference, requiring only a constant number of generation steps, using a Langevin dynamics-like sampler to generate the output beginning from Gaussian noise.

The key contributions of this paper are summarized as follows:

- WaveGrad combines recent techniques from score matching (Song et al., 2020; Song & Ermon, 2020) and diffusion probabilistic models (Sohl-Dickstein et al., 2015; Ho et al., 2020) to address conditional speech synthesis.
- We build and compare two variants of the WaveGrad model: (1) WaveGrad conditioned on a discrete refinement step index following Ho et al. (2020), (2) WaveGrad conditioned on a continuous scalar indicating the noise level. We find this novel continuous variant is more effective, especially because once the model is trained, different number of refinement steps can be used for inference. The proposed continuous noise schedule enables our model to use fewer inference iterations while maintaining the same quality (e.g., 6 vs. 50).
- We demonstrate that WaveGrad is capable of generating high fidelity audio samples, outperforming adversarial non-autoregressive models (Yamamoto et al., 2020; Kumar et al., 2019; Yang et al., 2020a; Bińkowski et al., 2020) and matching one of the best autoregressive models (Kalchbrenner et al., 2018) in terms of subjective naturalness. WaveGrad is capable of generating high fidelity samples using as few as six refinement steps.

## 2 ESTIMATING GRADIENTS FOR WAVEFORM GENERATION

We begin with a brief review of the Stein score function, Langevin dynamics, and score matching. The Stein score function (Hyvärinen, 2005) is the gradient of the data log-density $\log p(y)$ with respect to the datapoint $y$:

$$s(y) = \nabla_y \log p(y). \tag{1}$$

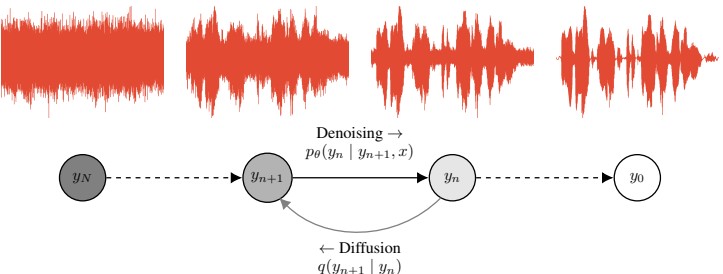

Figure 2: WaveGrad directed graphical model for training, conditioned on iteration index. $q(y_{n+1}|y_n)$ iteratively adds Gaussian noise to the signal starting from the waveform $y_0$. $q(y_{n+1}|y_0)$ is the noise distribution used for training. The inference denoising process progressively removes noise, starting from Gaussian noise $y_N$, akin to Langevin dynamics. Adapted from Ho et al. (2020).

Given the Stein score function $s(\cdot)$, one can draw samples from the corresponding density, $\tilde{y} \sim p(y)$, via Langevin dynamics, which can be interpreted as stochastic gradient ascent in the data space:

$$\tilde{y}_{i+1} = \tilde{y}_i + \frac{\eta}{2}s(\tilde{y}_i) + \sqrt{\eta}\,z_i, \tag{2}$$

where $\eta > 0$ is the step size, $z_i \sim \mathcal{N}(0, I)$, and $I$ denotes an identity matrix. A variant (Ho et al., 2020) is used as our inference procedure.

A generative model can be built by training a neural network to learn the Stein score function directly, using Langevin dynamics for inference. This approach, known as score matching (Hyvärinen, 2005; Vincent, 2011), has seen success in image (Song & Ermon, 2019; 2020) and shape (Cai et al., 2020) generation. The denoising score matching objective (Vincent, 2011) takes the form:

$$\mathbb{E}_{y \sim p(y)} \mathbb{E}_{\tilde{y} \sim q(\tilde{y}|y)} \left[ \left\| s_\theta(\tilde{y}) - \nabla_{\tilde{y}} \log q(\tilde{y} \mid y) \right\|_2^2 \right], \tag{3}$$

where $p(\cdot)$ is the data distribution, and $q(\cdot)$ is a noise distribution.

Recently, Song & Ermon (2019) proposed a weighted denoising score matching objective, in which data is perturbed with different levels of Gaussian noise, and the score function $s_\theta(\tilde{y}, \sigma)$ is conditioned on $\sigma$, the standard deviation of the noise used:

$$\sum_{\sigma \in S} \lambda(\sigma) \mathbb{E}_{y \sim p(y)} \mathbb{E}_{\tilde{y} \sim \mathcal{N}(y, \sigma)} \left[ \left\| s_\theta(\tilde{y}, \sigma) + \frac{\tilde{y} - y}{\sigma^2} \right\|_2^2 \right], \tag{4}$$

where $S$ is a set of standard deviation values that are used to perturb the data, and $\lambda(\sigma)$ is a weighting function for different $\sigma$. WaveGrad is a variant of this approach applied to learning *conditional* generative models of the form $p(y \mid x)$. WaveGrad adopts a similar objective which combines the idea of Vincent (2011); Ho et al. (2020); Song & Ermon (2019). WaveGrad learns the gradient of the data density, and uses a sampler similar to Langevin dynamics for inference.

The denoising score matching framework relies on a noise distribution to provide support for learning the gradient of the data log density (i.e., $q$ in Equation 3, and $\mathcal{N}(\cdot, \sigma)$ in Equation 4). The choice of the noise distribution is critical for achieving high quality samples (Song & Ermon, 2020). As shown in Figure 2, WaveGrad relies on the diffusion model framework (Sohl-Dickstein et al., 2015; Ho et al., 2020) to generate the noise distribution used to learn the score function.

## 2.1 WAVEGRAD AS A DIFFUSION PROBABILISTIC MODEL

Ho et al. (2020) observed that diffusion probabilistic models (Sohl-Dickstein et al., 2015) and score matching objectives (Song & Ermon, 2019; Vincent, 2011; Song & Ermon, 2020) are closely related. As such, we will first introduce WaveGrad as a diffusion probabilistic model.

We adapt the diffusion model setup in Ho et al. (2020), from unconditional image generation to conditional raw audio waveform generation. WaveGrad models the conditional distribution $p_\theta(y_0 \mid$

| **Algorithm 1** Training. WaveGrad directly conditions on the continuous noise level $\sqrt{\bar\alpha}$. $l$ is from a predefined noise schedule. | **Algorithm 2** Sampling. WaveGrad generates samples following a gradient-based sampler similar to Langevin dynamics. |
|---|---|
| 1: **repeat** 
 2: $\quad y_0 \sim q(y_0)$ 
 3: $\quad s \sim \text{Uniform}(\{1, \dots, S\})$ 
 4: $\quad \sqrt{\bar\alpha} \sim \text{Uniform}(l_{s-1}, l_s)$ 
 5: $\quad \epsilon \sim \mathcal{N}(0, I)$ 
 6: $\quad$ Take gradient descent step on 
 $\quad\quad \nabla_\theta \big\| \epsilon - \epsilon_\theta(\sqrt{\bar\alpha}\, y_0 + \sqrt{1 - \bar\alpha}\, \epsilon, x, \sqrt{\bar\alpha}) \big\|_1$ 
 7: **until** converged | 1: $y_N \sim \mathcal{N}(0, I)$ 
 2: **for** $n = N, \dots, 1$ **do** 
 3: $\quad z \sim \mathcal{N}(0, I)$ 
 4: $\quad y_{n-1} = \dfrac{\left( y_n - \frac{1-\alpha_n}{\sqrt{1-\bar\alpha_n}}\, \epsilon_\theta(y_n, x, \sqrt{\bar\alpha_n}) \right)}{\sqrt{\alpha_n}}$ 
 5: $\quad$ if $n > 1$, $y_{n-1} = y_{n-1} + \sigma_n z$ 
 6: **end for** 
 7: **return** $y_0$ |

$x$) where $y_0$ is the waveform and $x$ contains the conditioning features corresponding to $y_0$, such as linguistic features derived from the corresponding text, mel-spectrogram features extracted from $y_0$, or acoustic features predicted by a Tacotron-style text-to-speech synthesis model (Shen et al., 2018):

$$p_\theta(y_0 \mid x) := \int p_\theta(y_{0:N} \mid x)\, \mathrm{d}y_{1:N}, \tag{5}$$

where $y_1, \dots, y_N$ is a series of latent variables, each of which are of the same dimension as the data $y_0$, and $N$ is the number of latent variables (iterations). The posterior $q(y_{1:N} \mid y_0)$ is called the diffusion process (or forward process), and is defined through the Markov chain:

$$q(y_{1:N} \mid y_0) := \prod_{n=1}^{N} q(y_n \mid y_{n-1}), \tag{6}$$

where each iteration adds Gaussian noise:

$$q(y_n \mid y_{n-1}) := \mathcal{N}\left(y_n; \sqrt{(1 - \beta_n)}\, y_{n-1}, \beta_n I\right), \tag{7}$$

under some (fixed constant) noise schedule $\beta_1, \dots, \beta_N$. We emphasize the property observed by Ho et al. (2020), the diffusion process can be computed for any step $n$ in a closed form:

$$y_n = \sqrt{\bar\alpha_n}\, y_0 + \sqrt{(1 - \bar\alpha_n)}\, \epsilon \tag{8}$$

where $\epsilon \sim \mathcal{N}(0, I)$, $\alpha_n := 1 - \beta_n$ and $\bar\alpha_n := \prod_{s=1}^{n} \alpha_s$. The gradient of this noise distribution is

$$\nabla_{y_n} \log q(y_n \mid y_0) = -\frac{\epsilon}{\sqrt{1 - \bar\alpha_n}}. \tag{9}$$

Ho et al. (2020) proposed to train on pairs $(y_0, y_n)$, and to reparameterize the neural network to model $\epsilon_\theta$. This objective resembles denoising score matching as in Equation 3 (Vincent, 2011):

$$\mathbb{E}_{n,\epsilon}\left[ C_n \left\| \epsilon_\theta\left(\sqrt{\bar\alpha_n}\, y_0 + \sqrt{1 - \bar\alpha_n}\, \epsilon, x, n\right) - \epsilon \right\|_2^2 \right], \tag{10}$$

where $C_n$ is a constant related to $\beta_n$. In practice Ho et al. (2020) found it beneficial to drop the $C_n$ term, resulting in a weighted variational lower bound of the log-likelihood. Additionally in Ho et al. (2020), $\epsilon_\theta$ conditions on the discrete index $n$, as we will discuss further below. We also found that substituting the original $L_2$ distance metric with $L_1$ offers better training stability.

## 2.2 Noise Schedule and Conditioning on Noise Level

In the score matching setup, Song & Ermon (2019; 2020) noted the importance of the choice of noise distribution used during training, since it provides support for modelling the gradient distribution. The diffusion framework can be viewed as a specific approach to providing support to score matching, where the noise schedule is parameterized by $\beta_1, \dots, \beta_N$, as described in the previous section. This is typically determined via some hyperparameter heuristic, e.g., a linear decay schedule (Ho et al., 2020). We found the choice of the noise schedule to be critical towards achieving high fidelity audio in our experiments, especially when trying to minimize the number of inference iterations $N$ to make inference efficient. A schedule with superfluous noise may result in a model

unable to recover the low amplitude detail of the waveform, while a schedule with too little noise may result in a model that converges poorly during inference. Song & Ermon (2020) provide some insights around tuning the noise schedule under the score matching framework. We will connect some of these insights and apply them to WaveGrad under the diffusion framework.

Another closely related problem is determining $N$, the number of diffusion/denoising steps. A large $N$ equips the model with more computational capacity, and may improve sample quality. However using a small $N$ results in faster inference and lower computational costs. Song & Ermon (2019) used $N = 10$ to generate $32 \times 32$ images, while Ho et al. (2020) used 1,000 iterations to generate high resolution $256 \times 256$ images. In our case, WaveGrad generates audio sampled at 24 kHz.

We found that tuning both the noise schedule and $N$ in conjunction was critical to attaining high fidelity audio, especially when $N$ is small. If these hyperparameters are poorly tuned, the training sampling procedure may provide deficient support for the distribution. Consequently, during inference, the sampler may converge poorly when the sampling trajectory encounters regions that deviate from the conditions seen during training. However, tuning these hyperparameters can be costly due to the large search space, as a large number of models need to be trained and evaluated. We make empirical observations and discuss this in more details in Section 4.4.

We address some of the issues above in our WaveGrad implementation. First, compared to the diffusion probabilistic model from Ho et al. (2020), we reparameterize the model to condition on the continuous noise level $\bar{\alpha}$ instead of the discrete iteration index $n$. The loss becomes

$$\mathbb{E}_{\bar{\alpha}, \epsilon}\left[\left\|\epsilon_\theta\left(\sqrt{\bar{\alpha}}\, y_0 + \sqrt{1-\bar{\alpha}}\, \epsilon, x, \sqrt{\bar{\alpha}}\right) - \epsilon\right\|_1\right], \tag{11}$$

A similar approach was also used in the score matching framework (Song & Ermon, 2019; 2020), wherein they conditioned on the noise variance.

There is one minor technical issue we must resolve in this approach. In the diffusion probabilistic model training procedure conditioned on the discrete iteration index (Equation 10), we would sample $n \sim \mathrm{Uniform}(\{1, \ldots, N\})$, and then compute its corresponding $\alpha_n$. When directly conditioning on the continuous noise level, we need to define a sampling procedure that can directly sample $\bar{\alpha}$. Recall that $\bar{\alpha}_n := \prod_s^n (1 - \beta_s) \in [0, 1]$. While we could simply sample from the uniform distribution $\bar{\alpha} \sim \mathrm{Uniform}(0, 1)$, we found this to give poor empirical results. Instead, we use a simple hierarchical sampling method that mimics the discrete sampling strategy. We first define a noise schedule with $S$ iterations and compute all of its corresponding $\sqrt{\bar{\alpha}_s}$:

$$l_0 = 1, \qquad l_s = \sqrt{\prod_{i=1}^{s}(1 - \beta_i)}. \tag{12}$$

We first sample a segment $s \sim U(\{1, \ldots, S\})$, which provides a segment $(l_{s-1}, l_s)$, and then sample from this segment uniformly to give $\sqrt{\bar{\alpha}}$. The full WaveGrad training algorithm using this sampling procedure is illustrated in Algorithm 1.

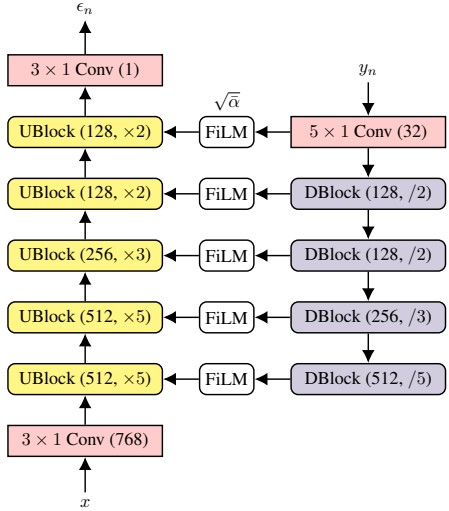

One benefit of this variant is that the model needs to be trained only once, yet inference can be run over a large space of trajectories without the need to be retrained. To be specific, once we train a model, we can use a different number of iterations $N$ during inference, making it possible to explicitly trade off between inference computation and output quality in one model. This also makes fast hyperparameter search possible, as we will illustrate in Section 4.4. The full inference algorithm is explained in Algorithm 2. The full WaveGrad architecture is visualized in Figure 3. Details are included in Appendix A.

## 3 RELATED WORK

This work is inspired in part by Sohl-Dickstein et al. (2015), which applies diffusion proba-

Figure 3: WaveGrad network architecture. The inputs consists of the mel-spectrogram conditioning signal $x$, the noisy waveform generated from the previous iteration $y_n$, and the noise level $\sqrt{\bar{\alpha}}$. The model produces $\epsilon_n$ at each iteration, which can be interpreted as the direction to update $y_n$.

bilistic models to unconditional image synthesis, whereas we apply diffusion probabilistic models to conditional generation of waveform. The objective we use also resembles the Noise Conditional Score Networks (NCSN) objective of Song & Ermon (2019). Similar to Song & Ermon (2019; 2020), our models condition on a continuous scalar indicating the noise level. Denoising score matching (Vincent, 2011) and sliced score matching Song et al. (2020) also use similar objective functions, however they do not condition on the noise level. Work of Saremi et al. (2018) on score matching is also related, in that their objective accounts for a noise hyperparameter. Finally, Cai et al. (2020) applied NCSN to model conditional distributions for shape generation, while our focus is waveform generation.

WaveGrad also closely relates to masked-based generative models (Devlin et al., 2019; Lee et al., 2018; Ghazvininejad et al., 2019; Chan et al., 2020; Saharia et al., 2020), insertion-based generative models (Stern et al., 2019; Chan et al., 2019b;a;c; Li & Chan, 2019) and edit-based generative models (Sabour et al., 2019; Gu et al., 2019; Ruis et al., 2019) found in the semi-autoregressive sequence generation literature. These approaches model discrete tokens and use edit operations (e.g., insertion, substitution, deletion), whereas in our work, we model the (continuous) gradients in a continuous output space. Edit-based models can also iteratively refine the outputs during inference (Lee et al., 2018; Ghazvininejad et al., 2019; Chan et al., 2020), while they do not rely on a (continuous) gradient-based sampler, they rely on a (discrete) edit-based sampler. The noise distribution play a key role, token masking based on Bernoulli (Devlin et al., 2019), uniform (Saharia et al., 2020), or hand-crafted (Chan et al., 2020) distributions have been used to enable learning the edit distribution. We rely on a Markov chain from the diffusion framework (Ho et al., 2020) to sample perturbations.

We note that the concurrent work of Kong et al. (2020) also applies the diffusion framework of Ho et al. (2020) to waveform generation. Their model conditions on a discrete iteration index whereas we find that conditioning on a continuous noise level offers improved flexibility and enables generating high fidelity audio as few as six refinement steps. By contrast, Kong et al. (2020) report performance using 20 refinement steps and evaluate their models when conditioned on ground truth mel-spectrogram. We evaluate WaveGrad when conditioned on Tacotron 2 mel-spectrogram predictions, which corresponds to a more realistic TTS setting.

The neural network architecture of WaveGrad is heavily inspired by GAN-TTS (Bińkowski et al., 2020). The upsampling block (UBlock) of WaveGrad follows the GAN-TTS generator, with a minor difference that no BatchNorm is used.

## 4 EXPERIMENTS

We compare WaveGrad with other neural vocoders and carry out ablations using different noise schedules. We find that WaveGrad achieves the same sample quality as the fully autoregressive state-of-the-art model of Kalchbrenner et al. (2018) (WaveRNN) on both internal datasets (Table 1) and LJ Speech (Ito & Johnson, 2017) (Table C.1) with less sequential operations.

### 4.1 MODEL AND TRAINING SETUP

We trained models using a proprietary speech dataset consisted of 385 hours of high quality English speech from 84 professional voice talents. For evaluation, we chose a female speaker in the training dataset. Speech signals were downsampled to 24 kHz then 128-dimensional mel-spectrogram features (50 ms Hanning window, 12.5 ms frame shift, 2048-point FFT, 20 Hz & 12 kHz lower & upper frequency cutoffs) were extracted. During training, mel-spectrograms computed from ground truth audio were used as the conditioning features $x$. However, during inference, we used predicted mel-spectrograms generated by a Tacotron 2 model (Shen et al., 2018) as the conditioning signal. Although there was a mismatch in the conditioning signals between training and inference, unlike Shen et al. (2018), preliminary experiments demonstrated that training using ground truth mel-spectrograms as conditioning had no regression compared to training using predicted features. This property is highly beneficial as it significantly simplifies the training process of text-to-speech

models: the WaveGrad vocoder model can be trained separately on a large corpus without relying on a pretrained text-to-spectrogram model.

**Model Size:** Two network size variations were compared: Base and Large. The WaveGrad Base model took 24 frames corresponding to 0.3 seconds of audio (7,200 samples) as input during training. We set the batch size to 256. Models were trained on using 32 Tensor Processing Unit (TPU) v2 cores. The WaveGrad Base model contained 15M parameters. For the WaveGrad Large model, we repeated each UBlock/DBlock twice, one with upsampling/downsampling and another without. Each training sample included 60 frames corresponding to a 0.75 second of audio (18,000 samples). We used the same batch size and trained the model using 128 TPU v3 cores. The WaveGrad Large model contained 23M parameters. Both Base and Large models were trained for about 1M steps. The network architecture is fully convolutional and non-autoregressive thus it is highly parallelizable at both training and inference.

**Noise Schedule:** All noise schedules we used can be found in Appendix B.

## 4.2 EVALUATION

The following models were used as baselines in this experiment: (1) WaveRNN (Kalchbrenner et al., 2018) conditioned on mel-spectrograms predicted by a Tacotron 2 model in teacher-forcing mode following Shen et al. (2018); The model used a single long short-term memory (LSTM) layer with 1,024 hidden units, 5 convolutional layers with 512 channels as the conditioning stack to process the mel-spectrogram features, and a 10-component mixture of logistic distributions (Salimans et al., 2017) as its output layer, generating 16-bit samples at 24 kHz. It had 18M parameters and was trained for 1M steps. Preliminary experiments indicated that further reducing the number of units in the LSTM layer hurts performance. (2) Parallel WaveGAN (Yamamoto et al., 2020) with 1.57M parameters, trained for 1M steps. (3) MelGAN (Kumar et al., 2019) with 3.22M parameters, trained for 4M steps. (4) Multi-band MelGAN (Yang et al., 2020a) with 2.27M parameters, trained for 1M steps. (5) GAN-TTS (Bińkowski et al., 2020) with 21.4M parameters, trained for 1M steps.

All models were trained using the same training set as the WaveGrad models. Following the original papers, Parallel WaveGAN, MelGAN, and Multi-band MelGAN were conditioned on the mel-spectrograms computed from ground truth audio during training. They were trained using a publicly available implementation at `https://github.com/kan-bayashi/ParallelWaveGAN`. Note that hyper-parameters of these baseline models were not fully optimized for this dataset.

To compare these models, we report subjective listening test results rating speech naturalness on a 5-point Mean Opinion Score (MOS) scale, following the protocol described in Appendix D. Conditioning mel-spectrograms for the test set were predicted using a Tacotron 2 model, which were passed to these models to synthesize audio signals. Note that the Tacotron 2 model was identical to the one used to predict mel-spectrograms for training WaveRNN and GAN-TTS models.

## 4.3 RESULTS

Subjective evaluation results are summarized in Table 1. Models conditioned on discrete indices followed the formulation from Section 2.1, and models conditioned on continuous noise level followed the formulation from Section 2.2. WaveGrad models matched the performance of the autoregressive WaveRNN baseline and outperformed the non-autoregressive baselines. Although increasing the model size slightly improved naturalness, the difference was not statistically significant. The WaveGrad Base model using six iterations achieved a real time factor (RTF) of 0.2 on an NVIDIA V100 GPU, while still achieving an MOS above 4.4. As a comparison, the WaveRNN model achieved a RTF of 20.1 on the same GPU, 100 times slower. More detailed discussion is in Section 4.4. Appendix C contains results on a public dataset using the same model architecture and noise schedule.

## 4.4 DISCUSSION

To understand the impact of different noise schedules and to reduce the number of iterations in the noise schedule from 1,000, we explored different noise schedules using fewer iterations. We found that a well-behaved inference schedule should satisfy two conditions:

Table 1: Mean opinion scores (MOS) of various models and their confidence intervals. All models except WaveRNN are non-autoregressive. WaveGrad, Parallel WaveGAN, MelGAN, and Multi-band MelGAN were conditioned on the mel-spectrograms computed from ground truth audio during training. WaveRNN and GAN-TTS used predicted features for training.

| Model | MOS ($\uparrow$) |
| --- | --- |
| WaveRNN | $4.49 \pm 0.04$ |
| Parallel WaveGAN | $3.92 \pm 0.05$ |
| MelGAN | $3.95 \pm 0.06$ |
| Multi-band MelGAN | $4.10 \pm 0.05$ |
| GAN-TTS | $4.34 \pm 0.04$ |
| WaveGrad | |
|     Base (6 iterations, continuous noise levels) | $4.41 \pm 0.03$ |
|     Base (1,000 iterations, discrete indices) | $4.47 \pm 0.04$ |
|     Large (1,000 iterations, discrete indices) | $4.51 \pm 0.04$ |
| Ground Truth | $4.58 \pm 0.05$ |

1. The KL-divergence $D_{\mathrm{KL}}\left(q(y_N \mid y_0) \parallel \mathcal{N}(0, I)\right)$ between $y_N$ and standard normal distribution $\mathcal{N}(0, I)$ needs to be small. Large KL-divergence introduces mismatches between training and inference. To make the KL-divergence small, some $\beta$s need to be large enough.
2. $\beta$ should start with small values. This provides the model training with fine granularity details, which we found crucial for reducing background static noise.

In this section, all the experiments were conducted with the WaveGrad Base model. Both objective and subjective evaluation results are reported. The objective evaluation metrics include

1. Log-mel spectrogram mean squared error metrics (LS-MSE), computed using 50 ms window length and 6.25 ms frame shift;
2. Mel cepstral distance (MCD) (Kubichek, 1993), a similar MSE metric computed using 13-dimensional mel frequency cepstral coefficient features;
3. $F_0$ Frame Error (FFE) (Chu & Alwan, 2009), combining Gross Pitch Error and Voicing Decision metrics to measure the signal proportion whose estimated pitch differs from ground truth.

Since the ground truth waveform is required to compute objective evaluation metrics, we report results using ground truth mel-spectrograms as conditioning features. We used a validation set of 50 utterances for objective evaluation, including audio samples from multiple speakers. Note that for MOS evaluation, we used the same subjective evaluation protocol described in Appendix D. We experimented with different noise schedules and number of iterations. These models were trained with conditioning on the discrete index. Subjective and quantitative evaluation results are in Table 2.

We also performed a detailed study on the the WaveGrad model conditioned on the continuous noise level in the bottom part of Table 2. Compared to the model conditioned on the discrete index with a fixed training schedule (top of Table 2), conditioning on the continuous noise level generalized better, especially if the number of iterations was small. It can be seen from Table 2 that degradation with the model with six iterations was not significant. The model with six iterations achieved real time factor (RTF) = 0.2 on an NVIDIA V100 GPU and RTF = 1.5 on an Intel Xeon CPU (16 cores, 2.3GHz). As we did not optimize the inference code, further speed ups are likely possible.

## 5 CONCLUSION

In this paper, we presented WaveGrad, a novel conditional model for waveform generation which estimates the gradients of the data density, following the diffusion probabilistic model (Ho et al., 2020) and score matching framework (Song et al., 2020; Song & Ermon, 2020). WaveGrad starts from Gaussian white noise and iteratively updates the signal via a gradient-based sampler conditioned on the mel-spectrogram. WaveGrad is non-autoregressive, and requires only a constant number of generation steps during inference. We find that the model can generate high fidelity audio samples using as few as six iterations. WaveGrad is simple to train, and implicitly optimizes for the

Table 2: Objective and subjective metrics of the WaveGrad Base models. When conditioning on the discrete index, a separate model needs to be trained for each noise schedule. In contrast, a single model can be used with different noise schedules when conditioning on the noise level directly. This variant yields high fidelity samples using as few as six iterations.

| Iterations (schedule) | LS-MSE ($\downarrow$) | MCD ($\downarrow$) | FFE ($\downarrow$) | MOS ($\uparrow$) |
|---|---|---|---|---|
| **WaveGrad conditioned on discrete index** | | | | |
| 25 (Fibonacci) | 283 | 3.93 | 3.3% | $3.86 \pm 0.05$ |
| 50 (Linear $(1 \times 10^{-4}, 0.05)$) | 181 | 3.13 | 3.1% | $4.42 \pm 0.04$ |
| 1,000 (Linear $(1 \times 10^{-4}, 0.005)$) | 116 | 2.85 | 3.2% | $4.47 \pm 0.04$ |
| **WaveGrad conditioned on continuous noise level** | | | | |
| 6 (Manual) | 217 | 3.38 | 2.8% | $4.41 \pm 0.04$ |
| 25 (Fibonacci) | 185 | 3.33 | 2.8% | $4.44 \pm 0.04$ |
| 50 (Linear $(1 \times 10^{-4}, 0.05)$) | 177 | 3.23 | 2.7% | $4.43 \pm 0.04$ |
| 1,000 (Linear $(1 \times 10^{-4}, 0.005)$) | 106 | 2.85 | 3.0% | $4.46 \pm 0.03$ |

weighted variational lower-bound of the log-likelihood. The empirical experiments demonstrated WaveGrad to generate high fidelity audio samples matching a strong autoregressive baseline.

## AUTHOR CONTRIBUTIONS

Nanxin Chen wrote code, proposed the idea, ran all experiments and wrote the paper. Yu Zhang recruited collaborators, co-managed/advised the project, conducted evaluation, debugging model and editing paper. Heiga Zen helped conducting text-to-speech experiments and advised the project. Ron Weiss implemented the objective evaluation metrics and advised the project. Mohammad Norouzi suggested the use of denoising diffusion models for audio generation and helped with writing and advising the project. William Chan conceived the project, wrote code, wrote paper, and co-managed/advised the project.

## ACKNOWLEDGMENTS

The authors would like to thank Durk Kingma, Yang Song, Kevin Swersky and Yonghui Wu for providing insightful research discussions and feedback. We also would like to thank Norman Casagrande for helping us to include the GAN-TTS baseline.

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

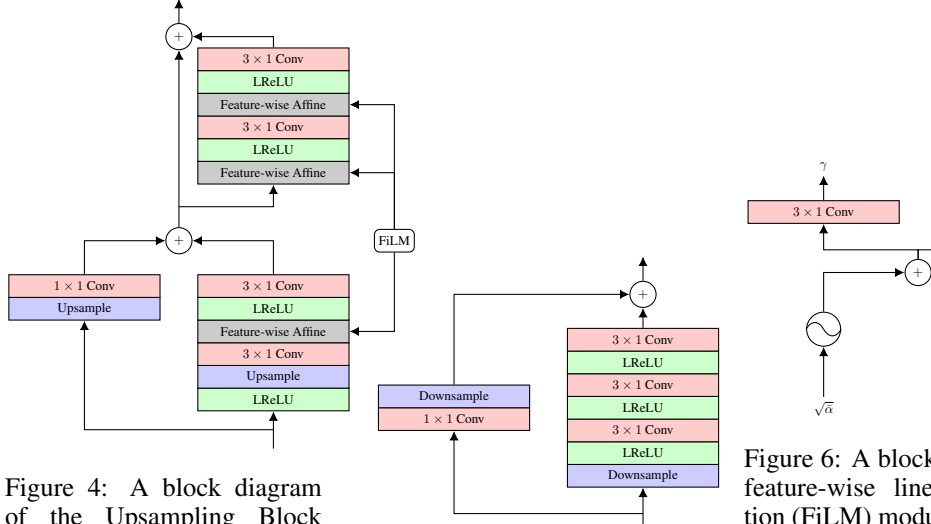

Figure 4: A block diagram of the Upsampling Block (UBlock). We upsample the signal modulated with information from the FiLM module.

Figure 5: A block diagram of the downsampling block (DBlock).

Figure 6: A block diagram of feature-wise linear modulation (FiLM) module. We condition on the noise level $\sqrt{\bar{\alpha}}$ of diffusion/denoising process, and pass it to a positional encoding function.

## A    NEURAL NETWORK ARCHITECTURE

To convert the mel-spectrogram signal (80 Hz) into raw audio (24 kHz), five upsampling blocks (UBlock) are applied to gradually upsample the temporal dimension by factors of 5, 5, 3, 2, 2, with the number of channels of 512, 512, 256, 128, 128 respectively. Additionally, one convolutional layer is added before and after these blocks.

The UBlock is illustrated in Figure 4. Each UBlock includes two residual blocks (He et al., 2016). Neural audio generation models often use large receptive field (Oord et al., 2016; Bińkowski et al., 2020; Yamamoto et al., 2020). The dilation factors of four convolutional layers are 1, 2, 4, 8 for the first three UBlocks and 1, 2, 1, 2 for the rest. Upsampling is carried out by repeating the nearest input. For the large model, we use 1, 2, 4, 8 for all UBlocks.

As an iterative approach, the network prediction is also conditioned on noisy waveform $\sqrt{\bar{\alpha}_n}\, y_0 + \sqrt{1 - \bar{\alpha}_n}\epsilon$. Downsampling blocks (DBlock), illustrated in Figure 5, are introduced to downsample the temporal dimension of the noisy waveform. The DBlock is similar to UBlock except that only one residual block is included. The dilation factors are 1, 2, 4 in the main branch. Downsampling is carried out by convolution with strides. Orthogonal initialization (Saxe et al., 2014) is used for all UBlocks and DBlocks.

The feature-wise linear modulation (FiLM) (Dumoulin et al., 2018) module combines information from both noisy waveform and input mel-spectrogram. We also represent the iteration index $n$, which indicates the noise level of the input waveform, using Transformer-style sinusoidal positional embeddings (Vaswani et al., 2017). To condition on the noise level directly, we also utilize the sinusoidal embeddings where $5000\sqrt{\bar{\alpha}}$ instead of $n$ is used. The FiLM module produces both scale and bias vectors given inputs, which are used in a UBlock for feature-wise affine transformation as

$$\gamma(D, \sqrt{\bar{\alpha}}) \odot U + \xi(D, \sqrt{\bar{\alpha}}), \tag{13}$$

where $\gamma$ and $\xi$ correspond to the scaling and shift vectors from the FiLM module, $D$ is the output from corresponding DBlock, $U$ is an intermediate output in the UBlock, and $\odot$ denotes the Hadamard product.

An overview of the FiLM module is illustrated in Figure 6. The structure is inspired by spatially-adaptive denormalization (Park et al., 2019). However batch normalization (Ioffe & Szegedy, 2015) is not applied in our work since each minibatch contains samples with different levels of noise. Batch statistics are not accurate since they are heavily dependent on sampled noise level. Experiment

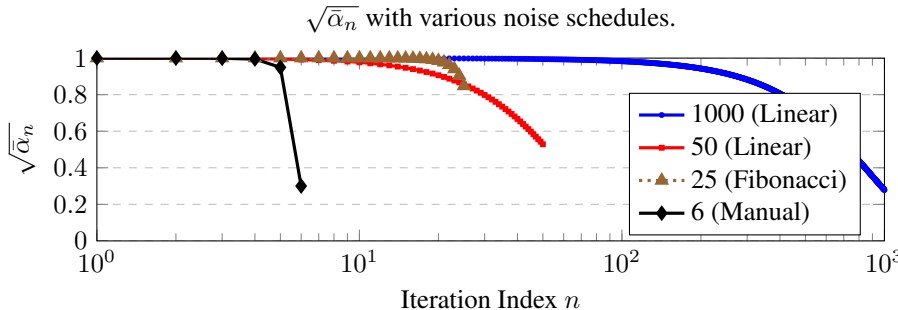

Figure 7: Plot of different noise schedules.

results also verified our assumption that models trained with batch normalization generate low-quality audio.

## B    NOISE SCHEDULE

For the WaveGrad Base model, we tested different noise schedules during training. For 1000 and 50 iterations, we set the forward process variances to constants increasing linearly from $\beta_1$ to $\beta_N$, defined as Linear($\beta_1$, $\beta_N$, $N$). We used Linear($1 \times 10^{-4}$, 0.005, 1000) for 1000 iterations and Linear($1 \times 10^{-4}$, 0.05, 50) for 50 iterations. For 25 iteration, a different Fibonacci-based schedule was adopted (referred to as Fibonacci($N$)):

$$\beta_0 = 1 \times 10^{-6} \quad \beta_1 = 2 \times 10^{-6}$$
$$\beta_n = \beta_{n-1} + \beta_{n-2} \quad \forall n \geq 2. \tag{14}$$

When a fixed schedule was used during training, the same schedule was used during inference. We found that a mismatch in the noise schedule degraded performance.

To sample the noise level $\sqrt{\bar{\alpha}}$, we set the maximal iteration $S$ to 1000 and precompute $l_1$ to $l_S$ from Linear($1 \times 10^{-6}$, 0.01, 1000). Unlike the base fixed schedule, WaveGrad support using a different schedule during inference thus "Manual" schedule was also explored to demonstrate the possibilities with WaveGrad. For example, the 6-iteration inference schedule was explored by sweeping the $\beta$s over following possibilities:

$$\{1, 2, 3, 4, 5, 6, 7, 8, 9\} \times 10^{-6}, 10^{-5}, 10^{-4}, 10^{-3}, 10^{-2}, 10^{-1} \tag{15}$$

Again, we did not need to train individual models for such hyper-parameter tuning. Here we used LS-MSE as a metric for tuning.

All noise schedules and corresponding $\sqrt{\bar{\alpha}}$ are plotted in Figure 7.

## C    RESULTS FOR LJ SPEECH

We ran experiments using the LJ Speech dataset (Ito & Johnson, 2017), a publicly available dataset consisting of audiobook recordings that were segmented into utterances of up to 10 seconds. We trained on a 12,764-utterance subset (23 hours) and evaluated on a held-out 130-utterance subset, following Battenberg et al. (2020). During training, mel-spectrograms computed from ground truth audio was used as the conditioning features. We used the held-out subset for evaluating synthesized speech with ground truth features. Results are presented in Table C.1. For this dataset, larger network size is beneficial and WaveGrad also matches the performance of the autoregressive baseline.

## D    SUBJECTIVE LISTENING TEST PROTOCOL

The test set included 1,000 sentences. Subjects were asked to rate the naturalness of each stimulus after listening to it. Following previous studies, a five-point Likert scale score (1: Bad, 2: Poor, 3:

Table C.1: Mean opinion scores (MOS) for LJ speech datasets.

| Model | MOS ($\uparrow$) |
|---|---|
| WaveRNN | $4.49 \pm 0.05$ |
| WaveGrad | |
|     Large (6 iterations, continuous noise levels) | $4.47 \pm 0.04$ |
|     Large (1000 iterations, continuous noise levels) | $4.55 \pm 0.05$ |
|     Base (6 iterations, continuous noise levels) | $4.35 \pm 0.05$ |
|     Base (1000 iterations, continuous noise levels) | $4.40 \pm 0.05$ |
| Ground Truth | $4.55 \pm 0.04$ |

Table E.2: Reported mean opinion scores (MOS) of various models and their confidence intervals. "Linguistic" and "Mel" in the "Features" column indicate that linguistic features and mel-spectrogram were used as conditioning, respectively.

| Model | Features | Sample Rate | MOS |
|---|---|---|---|
| Autoregressive | | | |
|     WaveNet (Oord et al., 2016) | Linguistic | 16 kHz | $4.21 \pm 0.08$ |
|     WaveNet (Oord et al., 2018) | Linguistic | 24 kHz | $4.41 \pm 0.07$ |
|     WaveNet (Shen et al., 2018) | Mel | 24 kHz | $4.53 \pm 0.07$ |
|     WaveRNN (Kalchbrenner et al., 2018) | Linguistic | 24 kHz | $4.46 \pm 0.07$ |
| Non-autoregressive | | | |
|     Parallel WaveNet (Oord et al., 2018) | Linguistic | 24 kHz | $4.41 \pm 0.08$ |
|     GAN-TTS (Bińkowski et al., 2020) | Linguistic | 24 kHz | $4.21 \pm 0.05$ |
|     GED (Gritsenko et al., 2020) | Linguistic | 24 kHz | $4.25 \pm 0.06$ |

Fair, 4: Good, 5: Excellent) was adopted with rating increments of 0.5. Each subject was allowed evaluate up to six stimuli. Test stimuli were randomly chosen and presented for each subject. Each stimulus was presented to a subject in isolation and was evaluated by one subject. The subjects were paid and native speakers of English living in United States. They were requested to use headphones in a quiet room.

# E    SUBJECTIVE SCORES REPORTED IN THE PRIOR WORK

Table E.2 shows the reported mean opinion scores of the prior work which used the same speaker. Although different papers listed here used the same female speaker, their results are not directly comparable due to differences in the training dataset, sampling rates, conditioning features, and sentences used for evaluation.

