# OpenReview forum: "WaveGrad: Estimating Gradients for Waveform Generation"
_ICLR.cc/2021/Conference — ICLR 2021 Poster_

### Official Review · AnonReviewer3 · 2020-10-28
**Official Blind Review #3**

**Rating:** 5
**Confidence:** 4

**Review:**

Summary:

The authors propose a conditional waveform synthesis model: WaveGrad, which combines recent methods from score matching and diffusion probabilistic models.

Pros:

(1) The proposed diffusion probabilistic model-based approach achieves comparable results as the autoregressive WaveRNN and outperformed the non-autoregressive baselines.

(2) The WaveGrad Base model using six iterations is much faster than the WaveRNN.

Cons:

(1) The authors borrow ideas from Ho et al. (2020) for image generation to address the conditional waveform synthesis problem.  There are really limited innovations on the model side since the proposed method just combines some techniques from recent score matching and diffusion probabilistic models.

(2) Why the proposed WaveGrad Vocoder can synthesize high-quality speech comparing to LSTMs in WaveRNN and WaveNet in the original Tacotron 2? The authors should give more insights and explanations to better illustrate the proposed method.

(3) The original Tacotron 2 with WaveNet can achieve 4.526 ± 0.066, which is better than the WaveGrad. The authors should also include the results of the WaveNet Vocoder.

(4) The authors should also provide results from the compared methods in https://wavegrad-iclr2021.github.io/ to validate the superiority of the proposed method.

*** Post-Rebuttal ***

Thank the authors for responding to my concerns in the rebuttal.

For the contribution part,  the major technical contribution is the continuous noise schedule. However, it is very obscure in the original paper. As also suggested by R1, the authors should carefully revise the paper to make it more clear. In addition, I found the paper largely borrowed content from Ho et al. (2020). From the framework figure and Algorithms to texts and equations, only limited modifications are made. After reading the paper, my first impression is that the paper just uses a new model on image synthesis to address speech synthesis. The work is not well motivated and largely copying another paper in the method section is not a professional way.

Hope the authors can modify the paper by adding the new clarifications and explanations in the rebuttal to improve the paper.

---

> ### Author Response · Authors · 2020-11-18
> **Thanks for your feedback (1/2)**
>
> First of all, many thanks for the detailed feedback. Below we address each of the individual questions.
>
> Q1. "There are really limited innovations on the model side since the proposed method just combines some techniques from recent score matching and diffusion probabilistic models."
>
> Our key contribution can be summarized as follows:
> (1) WaveGrad applies the diffusion-probabilistic model while conditioning on the continuous noise, this allows our model to use any noise inference schedule. Ho et a., 2020 [1] relies on a fixed discrete noise schedule. Our continuous noise schedule also licenses our model to use fewer inference iterations while maintaining the same quality (e.g., 6 vs. 1000). This also licenses the same model to run a dynamic number of inference iterations depending on the choice of sample quality desired.
> (2) We adapt the score matching framework previously only applied to unconditional image generation to conditional speech synthesis. We discuss our modelling contributions in depth in Section 4.
>
> Q2. "Why the proposed WaveGrad Vocoder can synthesize high-quality speech comparing to LSTMs in WaveRNN and WaveNet in the original Tacotron 2?"
>
> WaveRNN and WaveNet are autoregressive models that capture the output distribution directly. WaveGrad models the score function, which implicitly models the output distribution. Ho et al. showed the score matching objective can be seen as a lower-bound to the log-likelihood, suggesting that score matching models may in theory be capable of matching autoregressive models. Empirical experiments in our work and prior work (Ho et al [1], Song et al [2]) demonstrate the effectiveness of this framework. Additionally, due to the neural network architectures, autoregressive models may be limited to capturing left-only context, while non-autoregressive score matching iterative framework may be more likely to capture larger bidirectional context.
>
> During inference, autoregressive models are iterative models that require `n` steps. At each step, the output is generated without allowing for any correction in future iterations. Errors generated (e.g., generated from random sampling) at any point during the inference process can easily cascade and corrupt the signal. This is especially true since `n` is large in waveform generation, and autoregressive models were trained with teacher forcing. In contrast, iterative refinement methods like WaveGrad do not suffer from this cascading phenomena. WaveGrad iteratively refines the whole speech signal with each iteration, permitting it to correct for errors generated by the previous iteration.
>
> Q3. "The original Tacotron 2 with WaveNet can achieve 4.526 ± 0.066, which is better than the WaveGrad. The authors should also include the results of the WaveNet Vocoder."
>
> From our internal experiments, also observed in [3], WaveRNN achieves similar performance as WaveNet  on the vocoding task while the inference speed of WaveRNN is much faster in comparison. That is the reason why we only include the performance of WaveRNN, which is already comparable with ground truth utterances from Table 1.
>
> Q4. "The authors should also provide results from the compared methods in https://wavegrad-iclr2021.github.io/ to validate the superiority of the proposed method."
>
> Thank you for your suggestion. We are in the process of generating audio samples from our baseline models and once we get it, we will update our demo page as soon as possible.
>
> [1] Ho, J., Jain, A. and Abbeel, P., 2020. Denoising diffusion probabilistic models. Advances in Neural Information Processing Systems, 33.
>
> [2] Song, Y. and Ermon, S., 2019. Generative modeling by estimating gradients of the data distribution. In Advances in Neural Information Processing Systems (pp. 11918-11930).
>
> [3] Hsu, P.C., Wang, C.H., Liu, A.T. and Lee, H.Y., 2019. Towards Robust Neural Vocoding for Speech Generation: A Survey. arXiv preprint arXiv:1912.02461.

---

> ### Author Response · Authors · 2020-11-19
> **Thanks for your feedback (2/2)**
>
> We updated our demo page to include audio samples from few baseline models.

---

### Official Review · AnonReviewer2 · 2020-10-28
**Good paper, incline to accept**

**Rating:** 7
**Confidence:** 3

**Review:**

Motivated by recent works on score matching and diffusion probabilistic models, the authors presented a conditional, non-autoregressive model for waveform generation. The paper is well motivated and easy to follow.
The model starts from Gaussian white noise signal and iteratively refines the signal via Langevin dynamics-like sampler. The model is capable to generate high fidelity audio, outperforming a few adversarial non-autoregressive baselines even using only 6 iterations. Though the fundamental approaches of this paper have been applied in other domains, making the approach work well for waveform generation is relatively new and not easy.
The authors also had a deep dive on noise schedule, and proposed a variant of WaveGrad which conditioned on continuous noise level. The authors showed clear benefits of this variant over conditioning on discrete index, including using few iterations to generate high fidelity audio, being able to tradeoff between inference computation and output quality, fast hyper-parameter searching. However, one minor concern is that, if inference computation is not a bottleneck, when using linear schedule, for both 50/1000 iteration scenarios, WaveGrad conditioned on continuous noise level does not show clear benefits over conditioning on discrete index.

The other minor concern is regarding the model size. When comparing with adversarial counterparts, most models are with fewer parameters than the proposed WaveGrad. Though the authors claimed that model size has little effect on the performance of WaveGrad (according to Table 1), but WaveGrad does seem to clearly benefit from larger model according to Table C.1, the MOS for LJ speech datasets.

---

> ### Author Response · Authors · 2020-11-18
> **Thanks for your feedback**
>
> Thank you very much for your encouraging review. Below we have tried to address all of your specific concern:
>
> Concern 1. "if inference computation is not a bottleneck, when using linear schedule, for both 50/1000 iteration scenarios, WaveGrad conditioned on continuous noise level does not show clear benefits over conditioning on discrete index."
>
> Indeed, performance of the two approaches is comparable when the number of iterations is large.  The benefits of conditioning on continuous levels are in 1) its flexibility -- the model does not need to be retrained when choosing a noise schedule for generation, and 2) its strong performance when using very few iterations.
>
> Concern 2. "Though the authors claimed that model size has little effect on the performance of WaveGrad (according to Table 1), but WaveGrad does seem to clearly benefit from larger model according to Table C.1, the MOS for LJ speech datasets."
>
> Yes, we found model size to have little effect on our proprietary dataset, while LJ speech has some small benefits. We believe this to be dataset specific. We note that our proprietary dataset (and especially our test set) is more representative of real applications. Also model hyper-parameters were actually tuned on our proprietary dataset instead of LJ Speech.

---

### Official Review · AnonReviewer4 · 2020-10-28
**A very good paper studying diffusion probabilistic models for conditional audio synthesis task**

**Rating:** 8
**Confidence:** 4

**Review:**

The work uses diffusion probabilistic models for conditional speech synthesis tasks, specifically to convert mel-spectrogram to the raw audio waveform. Results from the proposed approach match the state-of-the-art WaveRNN model. The paper is very well-written and it is quite easy to follow. The study of the total number of diffusion steps and two different ways (continuous and discrete) ways to feed it in the network is very interesting. It is quite relevant and important for speech synthesis tasks. Using this, authors are able to find a 6-step inference procedure that yields very competitive performance to WaveRNN while still being computationally feasible.

Pros:
1. Great results for the neural vocoding task
2. Exhaustive study of the diffusion steps and how to feed them to the network is valuable and original.

Cons:
1. Study of the width/depth of the network can be more exhaustive.

*Further comments*: Authors mention in the conclusion ````Wavegrad is simple to train`. What makes the authors say so? It would be great to substantiate it with evidence/comparison?

---

> ### Author Response · Authors · 2020-11-18
> **Thanks for your feedback**
>
> Many thanks for the encouraging feedback. Below we address your comment:
>
> "Authors mention in the conclusion `Wavegrad is simple to train`. What makes the authors say so? It would be great to substantiate it with evidence/comparison?"
>
> Our conclusion comes from observations during experiments. There are mainly three different types of baselines covered in this paper:
> * WaveRNN and WaveNet: we found those autoregressive models take longer time to train
> * GAN-based: GAN-based approaches usually require additional losses, for example, losses on different resolutions or STFT-based losses. Also GAN training in general is not stable without proper hyperparameters.
> * Flow-based: Flow-based models require specialized architectures, since the network must be invertible and differentiable.  Furthermore, for efficient training and inference, flows are generally constrained to have efficient to compute Jacobian and inverses, as in coupling-based flows such as WaveGlow.
>
> In comparison, our work imposes minimal restrictions on the model architecture and we used a simple L1 loss to learn the score function instead of data likelihood used in autoregressive models. As discussed in the experiment section, our model converged very quickly.

---

### Official Review · AnonReviewer1 · 2020-11-03

**Rating:** 6
**Confidence:** 5

**Review:**

This work proposes a neural vocoder based on denoising diffusion probabilistic model. It matches autoregressive neural vocoder in terms audio fidelity, but only requires constant number of sequential steps at synthesis.

Detailed comments:

1, The first key contribution is not convincing. The proposed method is a direct application of Ho et al. (2020) for waveform synthesis. The combination/connection between score matching and diffusion probabilistic model is a contribution in Ho et al. (2020).

2, In general, flow-based models (e.g., WaveGlow, WaveFlow) can also match autoregressive model in terms of audio fidelity, while being much faster at synthesis (e.g., >20 times faster than real-time). What is the advantage of using WaveGrad as a neural vocoder? It provides comparable audio quality as flow-based models, but it is slower at synthesis (e.g., up to 5x faster than real-time). BTW, the authors may discuss WaveFlow (a SOTA flow-based model), which also introduces a trade-off between autoregressive and non-autoregressive models. It also needs constant number of sequential steps at synthesis.

3, Section 2 is not well written and self-contained. For example, the authors simply list Equation (2-4) and references without further explanation. Indeed, they are not closely related to the technical contribution of this paper. One may either introduce in more detail, or skip some content as this work focuses on a particular parameterization of diffusion probabilistic model introduced by Ho et al. (2020).

Pros:
- Solid technical contribution.
- Good empirical results.

Cons:
- It's unclear to me whether WaveGrad is a promising neural vocoder (see my comment 2).
- Paper writing can be improved.

I would like to raise my rating if my concerns are properly addressed.

---

> ### Author Response · Authors · 2020-11-18
> **Thanks for your feedback**
>
> Thank you for the comments and for taking the time to review our work. In what follows we have provided a detailed response to your concerns:
>
> Q1. "The proposed method is a direct application of Ho et al. (2020) for waveform synthesis. The combination/connection between score matching and diffusion probabilistic model is a contribution in Ho et al. (2020)."
>
> WaveGrad applies the diffusion-probabilistic model while conditioning on the continuous noise, this allows our model to use any noise inference schedule. Ho et a., 2020 relies on a fixed discrete noise schedule. Our continuous noise schedule also enables thelicenses our model to use fewer inference iterations while maintaining the same quality (e.g., 6 vs. 1000). This also licenses the same model to run a dynamic number of inference iterations depending on the choice of sample quality desired.
>
> Q2. "What is the advantage of using WaveGrad as a neural vocoder? It provides comparable audio quality as flow-based models, but it is slower at synthesis (e.g., up to 5x faster than real-time). BTW, the authors may discuss WaveFlow (a SOTA flow-based model), which also introduces a trade-off between autoregressive and non-autoregressive models. It also needs a constant number of sequential steps at synthesis."
>
> The advantage of WaveGrad is to build a single model which can dynamically balance the tradeoff between generation speed and performance by tuning the noise inference schedule. WaveGrad is capable of generating high-fidelity audio samples matching the state-of-the-art autoregressive models in terms of subjective naturalness. Meanwhile with the same model parameters, WaveGrad is able to use only a few iterations (e.g., 6) to support fast sampling, which still outperform other non-autoregressive baselines.
>
> We are in the process of reproducing WaveFlow. We are anticipating WaveGrad to outperform WaveFlow. However, there is concurrent work in the literature, DiffWave [1], that demonstrates their diffusion probabilistic model ( similar to ours), outperforms WaveFlow.
>
> Q3. "Section 2 is not well written and self-contained. For example, the authors simply list Equation (2-4) and references without further explanation. Indeed, they are not closely related to the technical contribution of this paper. One may either introduce in more detail, or skip some content  as this work focuses on a particular parameterization of diffusion probabilistic model introduced by Ho et al. (2020)."
>
> We have taken this to heart and improved the writing of Section 2. Indeed Equation 2-3 are not closely related to the technical contribution of this paper however we think that including them is beneficial to help understanding. So we update Section 2 to include further explanations. More specifically:
> * Equation 2 is related to algorithm 2 so we added another sentence: "A variant [2] is used as our inference procedure."
> * Equation 3 and equation 4: "WaveGrad adopts a similar objective which combines the idea of [2,3,4]."
>
> [1] Kong, Z., Ping, W., Huang, J., Zhao, K. and Catanzaro, B., 2020. DiffWave: A Versatile Diffusion Model for Audio Synthesis. arXiv preprint arXiv:2009.09761.
>
> [2] Ho, J., Jain, A. and Abbeel, P., 2020. Denoising diffusion probabilistic models. Advances in Neural Information Processing Systems, 33.
>
> [3] Vincent, P., 2011. A connection between score matching and denoising autoencoders. Neural computation, 23(7), pp.1661-1674.
>
> [4] Song, Y. and Ermon, S., 2019. Generative modeling by estimating gradients of the data distribution. In Advances in Neural Information Processing Systems (pp. 11918-11930).

---

> > ### Comment · AnonReviewer1 · 2020-11-23
> > **Thanks for your detailed response**
> >
> > I've read your response. It addresses most of my concerns. I would like to raise my score to 6.  Please update the paper accordingly.
> >
> > Some follow-up comments:
> > 1, If the continuous noise schedule is the major contribution here, then the authors should make it clear in the first key contribution. The current one is obscure.
> >
> > 2, It would be much better to include the model footprints for comparison. In general, quality, speed, and model footprint are three important dimensions to compare. It would be great to include WaveFlow for comparison, since it provides a trade off between these measures for exact likelihood models.

---

> > > ### Author Response · Authors · 2020-11-24
> > > **Thank you for your fast response (1/2)**
> > >
> > > Once again, many thanks for the detailed feedback and also for the fast response.
> > >
> > > Q1. If the continuous noise schedule is the major contribution here, then the authors should make it clear in the first key contribution. The current one is obscure.
> > >
> > > Thanks for the suggestion. We are working on the text to highlight this contribution.
> > >
> > > Q2. It would be much better to include the model footprints for comparison. In general, quality, speed, and model footprint are three important dimensions to compare. It would be great to include WaveFlow for comparison, since it provides a trade off between these measures for exact likelihood models.
> > >
> > > We updated our demo page to include audio samples from WaveGlow and WaveFlow. We used official implementations, standard hyperparameters and trained the model on the same proprietary dataset as the proposed WaveGrad. We trained WaveGlow for 300k steps and WaveFlow for 1.8 million steps with 4 NVIDIA V100 GPUs. Because of the time limitation, we couldn’t finish subjective evaluations but we never see those models give us high quality results even with longer training time compared to GAN-based models and WaveGrad.
> > >
> > > We believe another advantage of WaveGrad is that it can serve different applications with one network by tuning the inference schedule. In contrast, WaveFlow needs to train multiple networks with different sizes.

---

> > > ### Author Response · Authors · 2020-11-24
> > > **Thank you for your fast response (2/2)**
> > >
> > > Q1. "If the continuous noise schedule is the major contribution here, then the authors should make it clear in the first key contribution. The current one is obscure."
> > >
> > > Thanks for the suggestion. We updated the contribution part as follows:
> > >
> > > We build and compare two variants of the WaveGrad model: (1) WaveGrad conditioned on a discrete refinement step index **following [1]**, (2) WaveGrad conditioned on a continuous scalar indicating the noise level.
> > > We find that **this novel** continuous variant is more effective, especially because once the model is trained, different number of refinement steps can be used for inference.
> > > **The proposed continuous noise schedule also enables our model to use fewer inference iterations while maintaining the same quality (e.g., 6 vs. 50)**.
> > >
> > > [1] Ho, J., Jain, A. and Abbeel, P., 2020. Denoising diffusion probabilistic models. Advances in Neural Information Processing Systems, 33.

---

### Author Response · Authors · 2020-11-24
**Thanks for all feedback**

We thank all the reviewers for their feedback. We made the following changes to address the reviewer’s comments:

* We updated contribution part to highlight the novelty of the proposed continuous schedule
* We updated Section 2 to make it easier to follow
* We updated our demo page to include audio samples from our baselines models and WaveGlow/WaveFlow

---

### Decision · Program_Chairs · 2021-01-07
**Final Decision**

**Decision:**

Accept (Poster)

**Comment:**

Reviewers and myself agree that the contribution is clear, significant, and has enough originality. Hence, my recommendation is to ACCEPT the paper. As a brief summary, I highlight below some pros and cons that arose during the review and meta-review processes.

Pros:
- Solid technical contribution, specially the use of continuous noise levels.
- Clever application of diffusion/score-matching models to a new domain and task, with conditioning.
- Good empirical results, both objective and subjective.
- Listening samples provided.

Cons:
- Lack of formal comparison with flow-based vocoders.
- Potentially limited novelty.
- No official code available.

Note: Readers may also be interested in concurrent work https://openreview.net/forum?id=a-xFK8Ymz5J ("DiffWave: A Versatile Diffusion Model for Audio Synthesis").